# Proteomic Analysis of Novel Components of *Nemopilema nomurai* Jellyfish Venom: Deciphering the Mode of Action

**DOI:** 10.3390/toxins11030153

**Published:** 2019-03-08

**Authors:** Indu Choudhary, Du Hyeon Hwang, Hyunkyoung Lee, Won Duk Yoon, Jinho Chae, Chang Hoon Han, Seungshic Yum, Changkeun Kang, Euikyung Kim

**Affiliations:** 1College of Veterinary Medicine, Gyeongsang National University, Jinju 52828, Korea; induchoudhary2u@gmail.com (I.C.); pretty830128@naver.com (H.L.); ckkang@gnu.ac.kr (C.K.); 2Institute of Animal Medicine, Gyeongsang National University, Jinju 52828, Korea; pooh9922@hanmail.net; 3Southeast Sea Fisheries Research Institute, National Institute of Fisheries Science, Tongyeong 53085, Korea; wondukyoon@daum.net; 4Marine Environmental Research and Information Laboratory, B1101, 17 Gosan-ro 148beon-gil, Gunpo-si, Gyeonggi-do 15850, Korea; jinhochae@gmail.com; 5Headquarters for Marine Environment, National Fisheries Research & Development Institute, Shiran-ri, Gijang-eup, Gijang-gun, Busan 619-705, Korea; hanch@korea.kr; 6Ecological Risk Research Division, Korea Institute of Ocean Science and Technology (KIOST), Geoje 53201, Korea; syum@kiost.ac.kr; 7Faculty of Marine Environmental Science, University of Science and Technology (UST), Geoje 53201, Korea

**Keywords:** Jellyfish, *Nemopilema nomurai*, NnV, 2-DE, MALDI/TOF/MS

## Abstract

Nowadays, proliferation of jellyfish has become a severe matter in many coastal areas around the world. Jellyfish *Nemopilema nomurai* is one of the most perilous organisms and leads to significant deleterious outcomes such as harm to the fishery, damage the coastal equipment, and moreover, its envenomation can be hazardous to the victims. Till now, the components of *Nemopilema nomurai* venom (NnV) are unknown owing to scant transcriptomics and genomic data. In the current research, we have explored a proteomic approach to identify NnV components and their interrelation with pathological effects caused by the jellyfish sting. Altogether, 150 proteins were identified, comprising toxins and other distinct proteins that are substantial in nematocyst genesis and nematocyte growth by employing two-dimensional gel electrophoresis and matrix-assisted laser desorption/ionization time of flight mass spectrometry (MALDI/TOF/MS). The identified toxins are phospholipase A2, phospholipase D Li Sic Tox beta IDI, a serine protease, putative Kunitz-type serine protease inhibitor, disintegrin and metalloproteinase, hemolysin, leukotoxin, three finger toxin MALT0044C, allergens, venom prothrombin activator trocarin D, tripeptide Gsp 9.1, and along with other toxin proteins. These toxins are relatively well characterized in the venoms of other poisonous species to induce pathogenesis, hemolysis, inflammation, proteolysis, blood coagulation, cytolysis, hemorrhagic activity, and type 1 hypersensitivity, suggesting that these toxins in NnV can also cause similar deleterious consequences. Our proteomic works indicate that NnV protein profile represents valuable source which leads to better understanding the clinical features of the jellyfish stings. As one of the largest jellyfish in the world, *Nemopilema nomurai* sting is considered to be harmful to humans due to its potent toxicity. The identification and functional characterization of its venom components have been poorly described and are beyond our knowledge. Here is the first report demonstrating the methodical overview of NnV proteomics research, providing significant information to understand the mechanism of NnV envenomation. Our proteomics findings can provide a platform for novel protein discovery and development of practical ways to deal with jellyfish stings on human beings.

## 1. Introduction

Over the recent decades, there is a huge expansion of jellyfish blooms worldwide, which cause severe damage to the fishery and disturb marine ecosystem [1]. Since 1983, more than 2000 cases of *N. nomurai* jellyfish accidents have been reported in the coastal areas of China, Korea, and Japan, and life-threatening cases were also observed in humans [2]. *N. nomurai* known as the giant jellyfish is one of the most dangerous species belonging to the Phylum Cnidaria, being their diagnostic feature presence of stinging organelles called nematocysts, located mostly on the tentacles of jellyfish [3,4]. On sudden stimulation, nematocysts explosively discharge various venom constituents into the preys or victims [4]. Many scientists performed toxicological research on NnV, which includes cardiotoxic, hepatotoxic, hemolytic and cytotoxic biological activities [5,6,7,8]. Edema, itching, burning sensation, and small vesicles with erythematous eruption appeared at the site of contact on the victim’s body after NnV envenomation [9,10]. In the previous research, we have evaluated the cardiotoxic effect of NnV in H9c2 cells using a proteomic strategy [6]. Beside the toxicological and pharmacological importance of NnV, till now its venom composition has not been well defined. Classification and isolation of toxic proteins is a hard and laborious process.

In many venomous creatures such as snakes, spiders, scorpions, and cone snails, proteomic approaches have been auspiciously exploited to purify and characterize their venom components [11,12,13,14]. A few investigators have also attempted to fractionate and identify jellyfish venom toxins [15,16,17,18]. However, it is not so astonishing that jellyfish venom transcriptomics, and genomic data are rarely available, that, if any, may aid in the detection of bioactive venom components individually. For the present study, we have used the proteomic approach by utilizing two-dimensional gel electrophoresis, followed by matrix-assisted laser desorption/ionization time of flight mass spectrometry (MALDI/TOF/MS) and bioinformatics analyses to determine NnV composition that could explain life-threatening and undesirable consequences in human envenomation.

## 2. Results

### 2.1. Identification of N. nomurai Nematocyst Proteins by Proteomic Characterization

The identification and functional characterization of NnV components have been poorly characterized. Herein, we described the proteomic profile of the NnV. Our present study has successfully demonstrated the proteomic characterization of NnV by utilizing 2-DE and MALDI/TOF/MS. 2-DE gel electrophoresis, revealing the venom components ranging from PI 3–10 and molecular weights between 15–250 kDa, as shown in Figure 1. The scanned 2-DE image was marked with arrows and boundaries, generated by Progenesis Same Spots software (Nonlinear Dynamics, New Castle, UK), as shown in Figure 2. In this study, a total of 150 proteins identified from the nematocysts of NnV, including some toxins and another distinct type of proteins which are substantial in nematocyst and nematocyte generation (Table 1, Appendix A). Interestingly, the identified toxins from *N. nomurai* jellyfish have shown high 71 sequence similarity with those of other venomous and poisonous animals. Mainly, it is composed of Phospholipase A2, Phospholipase D Li Sic Tox beta IDI, Serine protease, Putative Kunitz-type Serine protease inhibitor, Disintigrin and Metalloproteinase, Hemolysin, Leukotoxin, Three-finger toxin MALT0044C, allergens, Venom prothrombin activator trocarin D, Tripeptide Gsp 9.1 and Cell death abnormality protein 1 from *Caenorhabditis elegans*. Several toxin proteins shared homology with proteins from microorganisms such as RTX-III toxin determinant A from serotype 2 from *Actinobacillus pleuropneumoniae*, Shiga-like toxin 1 subunit A from Enterobacteria phage, Leukotoxin from *Pasteurella haemolytica*, Cell death abnormality protein 12 from *Saccharomyces cerevisiae*, and Pro-apoptotic serine protease nma111 from *Neurospora crassa*. Interestingly, it also includes non-toxic proteins during NnV profiling, such as PCNA-interacting partner, Fukutin, Cell death abnormality protein 1, UvrABC system protein A, Division control protein 7, and OTU domain-containing protein 7B.

### 2.2. Modified Zymography Identify Metalloproteinase and PLA2 in NnV

Zymography assay was performed to identify the *N. nomurai* jellyfish venom proteolytic activity using various types of substrates such as gelatin, casein, and fibrin, as shown in Figure 3. Interestingly, *N. nomurai* jellyfish venom possesses higher gelatinolytic, caseinolytic, and fibrinolytic behavior. The gelatinolytic activities of NnV showed the greatest proteolytic activity as comparision to casein and fibrin zymography and displayed different protein banding pattern between 150–25 kDa. The majority of the gelatinolytic activities were inhibited in the presence of broad-spectrum metalloproteinase inhibitor (1,10-phenanthroline), confirming that metalloproteinase-like enzymes are present in *N. nomurai* jellyfish venom. Furthermore, casein zymography showed strong enzymatic activity, which could be evaluated in between the range of 70–20 kDa. In fibrin zymography, weaker fibrinolytic activity was observed at 70–25 kDa. The caseinolytic and fibrinolytic activities disappeared in the presence of 1,10-phenanthroline. Despite this, we have performed 2-DE zymography under the non-reducing condition to determine the proteolytic patterns, as shown in Figure 4.

### 2.3. Ontological Classification of Differentially Expressed Proteins

All identified proteins were subsequently classified into four ontologies according to their molecular functions, biological processes, cell components, and protein classes. In the molecular function classification, the most abundant protein category is of catalytic activity (46%) and binding (42%). Few proteins are associated with transporter activity (5%), receptor activity and structural molecular activity (3%) and translation regulator activity (1%). For the molecular function catalytic and binding activities designated the first rank, which can be related to the toxins components in NnV and hint towards strong toxicity. In the biological process category, the two significant groups include a cellular process (31%) and the metabolic process (28%). Followed by the response to stimulus (8%), developmental process (7%), biological regulation (5%), immune system process (5%), localization (5%), and multicellular organismal process (4%). The small number of proteins are assigned with biological adhesion (3%) and reproduction (1S%), as shown in Figure 5. According to cellular components, most of the proteins are assigned to the cell part (37%), followed by the organelles (20%) and macromolecular complex (19%). A minority of these proteins are localized in the membrane (10%), extracellular region (8%), extracellular matrix (4%), and (2%) cell junction. In the category of the protein class hydrolase represented the dominant class (31%). Followed by nucleic acid binding (17%), enzyme modulator (10%), transcription factor (7%), lyase (6%), cell adhesion molecule and transfer/carrier protein (5%), ligase (4%), signaling molecule (3%), and isomerase (2%), as shown in Figure 5.

## 3. Discussion

Over the last few decades, jellyfish envenomation has turned into a universal health issue and gave rise to approximately 150 million envenomation cases yearly [19]. Jellyfish stings display a wide range of clinical symptoms of severe pain, skin inflammation, dermatitis, nausea, emesis, cardiovascular, and respiratory distress [19]. In the current study, we have explored a proteomic approach to identify NnV components by performing two-dimensional gel electrophoresis, in-gel digestion, and MALDI/TOF/MS. In this study, a total of 150 proteins were identified from the nematocysts containing NnV. These proteins include toxins that might cause severe effects after envenomation and another distinct type of proteins.

Metalloproteases are well known for hemorrhagic activities such as fibrinolysis, apoptosis, inhibit platelet aggregation, interfere with blood coagulation, cause pro-inflammatory activity and deactivate blood serine protease inhibitors [20]. Our proteomics results identified various Metalloproteinases in the NnV, matched with Zinc-metalloproteinase-disintegrin-like or snake venom metalloproteinase from *Crotalus durissus* snake species, A Disintegrin and Metalloproteinase with thrombospondin motifs 10, A Disintegrin and Metalloproteinase with thrombospondin motifs 23 from *Mus musculus*, along with ATP-dependent Zinc Metalloprotease FtsH from *Oenococcus oeni*. In the present study, our proteomic analysis showed that metalloproteinase are the second most abundant components of NnV, comprising 21% of venom proteome, as shown in Figure 6. Further, 1-D zymography assay (Figure 3) and 2-DE zymography assay (Figure 4) determined the proteolytic activity of NnV, which was visualized as clear white bands or spots, respectively, on a blue background.

Moreover, in our previous study, we have found that the venoms of the scyphozoan jellyfish species including *Nemopilema nomurai*, *Cyanea nozakii*, *Rhopilema esculenta*, and *Aurelia aurita* contains various Metalloproteinases [21], which shows proteolytic activity and induced cytotoxicity in NIH 3T3 cells [21]. Therefore, Metalloproteinases in the venom of jellyfish *Nemopilema nomurai* might be responsible for swelling, inflammation, and dermonecrosis [22]. Previously, matrix Metalloproteinases from the venom of *Nemopilema nomurai* were also reported to induce dermal toxicity in both the in-vivo and in-vitro animal model [22]. In future, search and application of Metalloprotease inhibitor on jellyfish venom will be beneficial in the management of jellyfish envenomation and development of novel therapeutic source against venom toxicity.

Several phospholipases were also identified in NnV in current research, such as Phospholipase D LiSicTox-betaIDI, also known as LiRecDT5 homologous to PLA2 from *Loxosceles intermedia* spider venom, 85/88 kDa Calcium-independent Phospholipase A2 and Phospholipase A-2-activating protein from *Rattus norvegicus*, Calcium-independent Phospholipase A2-gamma from *Mus musculus*. Phospholipase D LiSicTox-betaIDI is the novel member of *loxoscele*s dermonecrotic toxin family, which has diverse biological activities such as dermonecrosis, increased vessel permeability, platelet aggregation, induce an inflammatory response, and cause mortality in animal models [23,24]. The presence of phospholipases in NnV can be corelated to pathogenesis, hemolysis, and other harmful consequences, for example, burning sensation and an erythematous eruption with small vesicles after proper provocation. Our proteomic analysis revealed that phospholipases is another abundant toxin protein constituting 12% in whole venom proteome (Figure 6).

Proteases constituted a vital portion of NnV counting 5% of the venom proteome (Figure 6). Serine proteases are well studied in various venom sources, typically in the snake, cone snail, scorpion, and spider venoms [25,26,27,28]. Serine proteases were also reported in the marine animals such as sea snake, stingrays, sponge and jellyfish [29,30,31]. Serine proteases mainly cause haemotoxicity by affecting the blood coagulation system and triggers fibrinolysis, platelet aggregation, and edema [32]. In this study, we have identified several serine proteases matched with snake venom serine protease Nikobin from *Vipera nikolskii*, serine protease HtrA-like from *Staphylococcus aureus* (strain NCTC 8325), Pro-apoptotic serine protease nma111 from *Neurospora crassa* (strain ATCC 24698/74-OR23-1A/CBS 708.71/DSM 1257/FGSC 987). Snake venom Serine Protease Nikobin can acts on the hemostasis system of the prey by affecting several physiological processes that include blood coagulation, fibrinolysis, and blood pressure [32,33]. Accordingly, serine proteases obtained from NnV might be responsible for pathogenesis during NnV envenomation.

Putative Kunitz-type Serine protease inhibitor is efficiently characterized in well-known venomous animals, for example snakes, spiders, scorpions, centipedes, cone snails, and sea anemone [34,35,36,37,38,39]. Kunitz-type Serine protease inhibitor plays a crucial role in various physiological activities namely blood coagulation, fibrinolysis, inflammation and ion channel blocking [37,38,39,40]. Kunitz-type Serine protease inhibitor had inhibitory activity against trypsin, chymotrypsin, and demonstrated its antifibrinolytic activity in snake venom [39,40,41]. Additionally, recent studies revealed that spider Kunitz-type Serine protease inhibitor exhibits inhibitory activity against trypsin, chymotrypsin, plasmin, and neutrophil elastase [42]. In the present study, our proteomic data showed that NnV was also comprised of such a Putative Kunitz-type Serine protease inhibitor, which might contribute to NnV toxicity.

Other toxin protein types present in NnV are homologous to the three-finger toxin MALT0044C, hemolysins, leukotoxins, major pollen allergen Lol p 5a, Allergen Mag, venom prothrombin activator trocarin D, RTX-III toxin determinant A from serotype 2, and Shiga-like toxin 1 subunit A. Three-finger toxin MALT0044C belongs to the snake three finger toxin family, it has a predominant role in neurotoxicity by inducing peripheral paralysis and finally leads to respiratory arrest and death [41]. The proteins in the 3FTx family have high functional diversity; they act as neurotoxins, which can target the cholinergic system and also block the L-type Ca^2+^ channels [42]. Therefore, three-finger toxin in NnV can also result in lethal health outcomes.

Various types of Hemolysins were reported in the venom of different kinds of jellyfish, including Ryncolin-2, Ryncolin-3, Neoverrucotoxin subunit beta, and Veficolin-1 were identified in the venom of jellyfish *S. meleagris* [43]. Moreover, CfTX-1 and CfTX-2 are potent Haemolysins were identified in *C. fleckeri* venom, which share sequence homology to toxins related to 4 cubozoan jellyfish species [44]. Other hemolytic proteins include CaTX-1 from *C. alata*, CrTX-1 from *C. rastoni*, and CcTX-1 from *C. capillata* were also reported in previous studies [45,46,47,48,49]. Former research indicated that all these hemolytic proteins are responsible for the hemolytic action and induce inflammation, pain, dermonecrosis, and death in animal models [45,46,47,48,49]. Presence of Hemolysins in NnV supports its previously reported evenomation symptoms like inflammation, pain, and dermonecrosis.

Hemolysin transporter protein was also identified in NnV, which is homologous to ShlB of *Serratia marcescens*. Hemolysin triggers the release of the inflammatory mediators, increase vascular permeability, cause edema formation, granulocyte accumulation, and finally contribute pathogenicity of Serratia species [50,51]. Therefore, it can be inferred that the Hemolysin present in NnV might impair hemostasis, and cause edema and hypotension after envenomation.

*Pasteurella* Leukotoxins are exotoxins that attack host leukocytes, chiefly polymorphonuclear cells, by inducing cell rupture [52]. The Leukotoxin binds to the host LFA-1 integrin and triggers a signaling cascade cause following biological effects such as tyrosine phosphorylation of the CD18 tail, increasing the intracellular Ca^2+^, finally results in host cell rupture [52]. We also identified Leukotoxins in the proteomics analysis of NnV homolog to *Pasteurella Haemo-lytica*-like sp. (strain 5943b), which may play a significant role in NnV envenomation and cause harmful consequences.

Two types of allergens were also detected in the NnV proteomics known as Major pollen allergen Lol p 5a from *Lolium perenne* and Allergen Mag (fragment) from *Dermatophagoides farina*. Lol p 5a is a major allergen of rye-grass pollen (*Lolium perenne pollen*), which mediate type I hypersensitivity, cause an allergic reaction in humans, hay fever, and triggers allergic asthma. House dust fly (*Dermatophagoides farina*) is the one of the major agent causing allergic ailment by way of bronchial, asthma, atopic dermatitis and rhinitis [53,54]. Allergen Mag is the major allergen found in *Dermatophagoides farina*, which binds to IgE and causes an allergic reaction in humans [55,56,57]. It releases histamine from washed blood cells of the mite-allergic patients. Hence, the allergens identified in NnV may be related to symptoms like dermatitis, utricularia, pruritus, and fever caused by jellyfish envenoming.

Venom prothrombin activator trocarin D is the snake prothrombin activator from *Tropidechis carinatus*, acts as a toxin component of the snake venom and disturbs the hemostatic system of prey [58]. It was reported that Venom prothrombin activator trocarin D induces cyanosis and death in mice at 1mg/kg body weight and display potent procoagulant effects [58]. Fortunately, we also identified the venom prothrombin activator trocarin D from NnV matched with that of *Tropidechis carinatus*. Therefore, venom prothrombin activator trocarin D is one of the important toxic protein in NnV which might cause severe harmful consequences such as shock, cardiovascular instability, and sudden death. RTX-III toxin determinant A from serotype 2 and RTX-I toxin determinant A from serotypes 1/9 are an essential virulence factor for *A. pleuropneumoniae* and cause swine pleuropneumonia [59].

RTX-III PROTEIN are toxic to porcine lung macrophages and erythrocytes, and it possesses strong hemolytic activity and causes cytotoxicity in alveolar macrophages and neutrophils hence play a significant role in pathogenesis [60]. Noteworthy, RTX-III toxin determinant A from serotype 2 was also found in NnV venom, and therefore, RTX-III toxin may account for hemolytic activity and cytotoxicity in jellyfish victims.

It was reported earlier that a Shiga-like toxin subunit A negatively regulates the translation process by catalytic inactivation of 60S ribosomal subunit and responsible for inhibiting protein synthesis [61]. The shiga-like toxin (SLT)-producing E. coli is associated with diarrhea, hemorrhagic colitis, and hemolytic uremic syndrome [61,62,63]. We also identified Shiga- like toxin subunit A in the proteomic analysis. So Shiga- like toxin subunit A could be responsible for cramps, abdominal pain, and other hemorrhagic activities after NnV stings.

Surprisingly, along with toxic proteins, we have also recognized many nontoxic proteins during NnV profiling, which plays an important role in signaling and other metabolic processes. Kinases itself constitute the major part of NnV, they hold 22% of known NnV proteins. Other proteins are 30% of total proteins found in NnV. OTU domain-containing protein 7B is one of the nontoxic proteins found in NnV. OTU domain-containing protein 7B is the negative regulator of the non-canonical NF-kappa-B pathway and displays the anticancer effect by suppressing the NF-kappa-B pathway in HCC cells [64]. OTU domain-containing protein 7B act as an anticancer target in liver cancer cells [64,65]. Hence NnV can exert an anticancer effect, and extensive studies need to investigate the therapeutic potential of NnV. Cell death abnormality protein 1, related to *Caenorhabditis elegans*, was also identified in NnV, and Cell death abnormality protein 1 necessitates the engulfment of cells undergoing programmed cell death [66,67]. It activates the expression of unfolded protein response genes and has a defense response to bacteria [68]. However, cell death abnormality protein 1 might play an essential role in nematocyst genesis and developments.

## 4. Conclusions

Jellyfish *Nemopilema nomurai* is one of the largest jellyfish and its evenomation can lead to harmful consequences, even death. Till now the identification and functional characterization of NnV components have been poorly described and remain beyond our knowledge. This is the first report that is able to identify around 150 proteins in *Nemopilema nomurai* jellyfish venom, which include metalloproteinases, kinases, phospholipases, proteases, toxins, and allergens. Some of these components are considered to play an important role in the cardiovascular, hepatotoxic, hematological, cytotoxic, and allergenic effects of the venom. These findings are beneficial in recognizing the mechanism of jellyfish envenomation and provide valuable means to establish new methods or treatments to handle jellyfish stings. The overall comprehensive characterization of NnV, provide a promising approach to explore the rich source of bioactive toxin components as therapeutic agents in the future.

## 5. Materials and Methods

### 5.1. Chemicals and Reagents

Immobiline™ Drystrip (pH 4–7, 18 cm), dithiothreitol (DTT), and iodoacetamide were bought from GE Healthcare life sciences (Marlborough, MA, USA). Acetonitrile (ACN), trifluoroacetic acid was purchased from Merck Chemicals, Darmstadt, Germany. Sequencing grade modified trypsin was purchased from Promega Corporation, Madison, WI, USA, formic acid from Acros Organics BVBA, Geel, Belgium. Other analytical grade chemicals and reagents were used and purchased from Sigma Aldrich Corporation (St. Louis, MO, USA).

### 5.2. Sample Collection and Preparation 

*N. nomurai* jellyfish specimens were collected from the Yellow Sea near the coast of Gunsan, South Korea. After taking out of water tentacles were dissected and transferred immediately in ice to the laboratory for further processing. Isolation of nematocysts was done using the previously described method [69]. In short, dissected tentacles were rinsed with cold seawater to eliminate any debris, and then 3 volumes (*v*/*v*) of cold sea water were added and at 4 °C put on shaker for one day. After that, the tentacle-free seawater was harvested and centrifuged at 1000× *g* for 5 min; then the nematocyst-rich pellet was washed thrice with fresh sea water. The remaining sediment tentacles were later autolyzed in fresh seawater at 4 °C for one day, as described above, and the autolysis process was repeated for 3–4 days. At last, the nematocysts that settled down were collected and seawater was used to wash them several times. Later, the nematocysts were centifuged at 500× *g* for 5 min, and pellets (nematocysts) were lyophilized and stored at −20 °C until further use.

### 5.3. Venom Extraction

The technique described by Carrette and Seymour was used to extract venom from the freeze-dried nematocysts [69] with slight modification. In short, venom was extracted from 50 mg of nematocyst using glass beads (approximately 8000 beads; 0.5 mm in diameter), and 1 mL of ice-cold phosphate buffered saline (PBS, pH 7.4). Further, samples were kept shaking on the mini bead mill at 3000 rpm for 30 s, the above step was repeated ten times with intermittent cooling on ice. The venom extracts were then transferred to a new Eppendorf tube and centrifuged (22,000 g) at 4 °C for 30 min. The above supernatant was used as NnV for the present study. Bradford assay (Bio-Rad, Hercules, CA, USA) [70] was used to determine the protein concentration of the venom. Finally, the venom was utilized in research, based on its protein concentration.

### 5.4. Two-Dimensional Gel Electrophoresis under Reducing Conditions and Image Analysis

For two-dimensional gel electrophoresis, 500 µg of protein samples were resuspended in 2-DE sample buffer (7 M urea, 2 M thiourea, 4% (*w*/*v*) CHAPS, 10 mg/mL DTT, 1% pharmalytes 3–10 and few grains of bromophenol blue). Then, protein samples were applied to immobiline TM Dry strip 18 cm, pH gradient (IPG) strips (pH 3–10) and passively rehydrated overnight at room temperature. The isoelectric focusing (IEF) was performed at a constant temperature of 20 °C using Ettan IPGphor system (GE Healthcare), with the following procedure: 50 V for 1:00 h, 200 V for 1:00 h, 500 V for 0:30 h, gradient 4000 V for 0:30 h, 4000 V for 1:00 h, gradient 10,000 V for 1:00 h, 10,000 V for 13:00 h, and 50 V for 3:00 h. Before the second dimension, the focused strips were firstly reduced with equilibration buffer (50 mM Tris-HCl (pH 8.8), 6 M urea, 30% glycerol, 2% SDS and 0.01% bromophenol blue containing 1% *w*/*v* DTT) for 15 min. For the second time, the gel strips were washed with the same equilibration buffer containing 2.5% *w*/*v* iodoacetamide for 15 min. The focused and equilibrated strips were inserted on top of the 12% SDS-PAGE and sealed with 0.5% *w*/*v* agarose gel. After that, electrophoresis was performed in a PROTEAN II xi cell gel electrophoresis unit. The gels were run at 10 mA/gel for 15 min for the initial migration and increased to 20 mA/gel at 20 °C for separation until the dye front reached the bottom of the gels. Silver staining was performed to visualize the protein spots as similar to the method reported by Mortz et al. [71], and this experiment was repeated thrice. The 2-DE gel images were digitalized by using Epson perfection V 700 photo scanner (Epson, Long Beach, CA, USA), and the acquired images were analyzed using Progenesis Same Spots software (Nonlinear Dynamics, Newcastle, UK).

### 5.5. In-gel Digestion

In-gel digestion of the proteins was performed by the method described previously [72]. The protein spots of interest were manually excised from preparative gels to perform in-gel digestion. In brief, the gel pieces were washed with pure water for several times and then destained with 30 mM potassium ferricyanide and 100 mM Na_2_S_2_O_3_ (50%/50% *v*/*v*) for 10 min with shaking. Further, the gel pieces were dehydrated for 10 min by incubating in 100 µl of 100% acetonitrile and were then dried in a lypholizer equipment for 15 min. The reduction step was performed by adding 50 µL reduction solution (10 mM DTT in 100 mM ammonium bicarbonate) to the dried gel pieces and incubated for 45 min at room temperature. After that, the gel pieces were alkylated with 100 mM iodoacetamide in the dark for 45 min at room temperature. The gel pieces were dried, incubated in trypsin (Promega, Madison, WI, USA) at a final concentration of 2 ng/µL in 10 µL of 50 mM NH_4_HCO_3_ and incubated on ice for 45 min. The excessive liquid was removed, and the proteolysis of protein was performed by adding the exact amount of 50 mM NH_4_HCO_3_. After overnight digestion with trypsin at 37 °C, the tryptic peptide mixture was pooled with extraction buffer containing 100% acetonitrile and 50% trifluoroacetic acid and concentrated in a speed vacuum.

### 5.6. MALDI/TOF/MS Analysis and Database Searching

The peptide samples were mixed with 1 µL of HCCAs matrix solution (α-acyano-4-hydroxycinnamic acid) and 1 µL of extraction buffer. 1 µL peptide mixture was spotted onto a freshly cleaned MALDI/TOF plate and dried at room temperature. Mass spectra were measured by using Voyager-DE STR mass spectrometer (Applied Biosystems, Franklin Lakes, NJ, USA).

Reflection/delayed extraction mode acquired the spectra. Monoisotopic peptide masses were chosen over a mass range of 800–3000 Da. Peptide mass fingerprinting (PMF) was performed to identify proteins using the MS-Fit program (http://prospector.ucsf.edu), and Mascot (Matrix science http://www.matrixscience.com) in the Swiss-Prot databases. For the peptide search following parameters were considered carbamidomethylation of cysteines as a fixed modification, oxidation of methionine as a variable modification, peptide mass tolerance of ±30 ppm for the fragment ions, trypsin with one missed cleavage was allowed. Criteria used for protein identification includes a number of the matched peptide, the extent of sequence coverage and probability based Mowse score was considered before accepting the identification.

### 5.7. Proteolytic Activity Assay

The proteolytic zymography assay was performed by using different types of substrates such as fibrin, gelatin, and casein, as the method described by References [73,74]. Respective zymography gels were prepared by copolymerized gelatin (2 mg/mL), casein (2 mg/mL), or fibrinogen (0.5 mg/mL)–thrombin (0.01 unit/mL), dissolved in 20 mM sodium phosphate buffer (pH 7.4) with 12% polyacrylamide. The protein samples were dissolved in non reducing sample buffer and the zymography gels were run under cold condition (4 °C) at 15 mA/gel. When the electrophoresis was over, the gels were washed with 2.5% Triton X-100 for 20 min to remove the SDS. After washing the gels were incubated with zymography reaction buffer (20 mM Tris (pH 7.4), 0.5 mM calcium chloride) at 37 °C overnight and stained with 0.125% coomassie blue. To study the protease inhibitor effect 1,10-phenanthroline was prepared as a 300 mM stock solution in methanol and added fresh to each incubation buffer to give a final concentration of 10 mM.

### 5.8. Modified Zymography Assays to Identify PLA2 Activity in NnV

NnV was run under non-reducing conditions on 12% SDS-PAGE and stained with Coomassie Brilliant Blue R-250. Blood and egg yolk zymography method [75] with slight modifications was to determine the molecular weight of PLA2. Briefly, to eradicate the SDS from the gel, the gel was washed for 1 h with 500 mM Tris-HCl (pH 7.4) containing 2% Triton X-100, and then for 1 h with 100 mM Tris-HCl (pH 7.4) along with 1% Triton X-100. Further gels were washed with 50 mM Tris-HCl (pH 7.4), 140 mM NaCl, and 2.5 mM CaCl_2_. For zymography on blood and egg-yolk, the washed SDS-PAGE gel was placed directly on a 1% agarose gel (50 mM Tris-HCl (pH 7.4), 140 mM NaCl, 2.5 mM CaCl_2_, 2.4% human erythrocytes, and 2% egg yolk. After overnight incubation at 37 °C, the opaque zone indicated the presence of PLA2 [76].

### 5.9. Two-Dimensional Gel Electrophoresis under Non-Reducing Conditions and 2-DE Gelatin Zymography

The NnV samples were prepared under non reducing conditions for 2-DE and gelatin 2-DE zymography. The 500 µg of protein samples were dissolved in 7 M urea, 2 M thiourea, 4% (*w*/*v*) CHAPS, 1% pharmalytes 3–10 and few grains of bromophenol blue) without DTT and the strips were rehydrated overnight, and the isoelectric focusing was performed as the method described above. After IEF the focused strips were washed with equilibration buffer (50 mM Tris-HCl (pH 8.8), 6 M urea, 30% glycerol, 2% SDS, and 0.01% bromophenol blue) for 15 min followed by electrophoresis. For 2-DE gelatin zymography 12% SDS-polyacrylamide gels were copolymerized with gelatin (1 mg/mL), and electrophoresis was performed at 4 °C. The gels were washed twice after electrophoresis for 30 min in 2.5% Triton X-100 to remove SDS. Triton X-100 traces were removed from the gels by washing with deionized water. After that, the gels were incubated in 20 mM Tris (pH 7.4), 0.5 mM calcium chloride at 37 °C for 16 h and stained with 0.125% coomassie blue. Clear zones of hydrolysis in the gel against blue background showed gelatin proteolytic activity.

### 5.10. Gene Ontology Analysis of the Identified Venom Proteins

The proteomic data was analyzed using the Panther classification system (http:www.pantherdb.org/) to classify regarding molecular function, biological process, protein class, and cellular components [77].

## Figures and Tables

**Figure 1 toxins-11-00153-f001:**
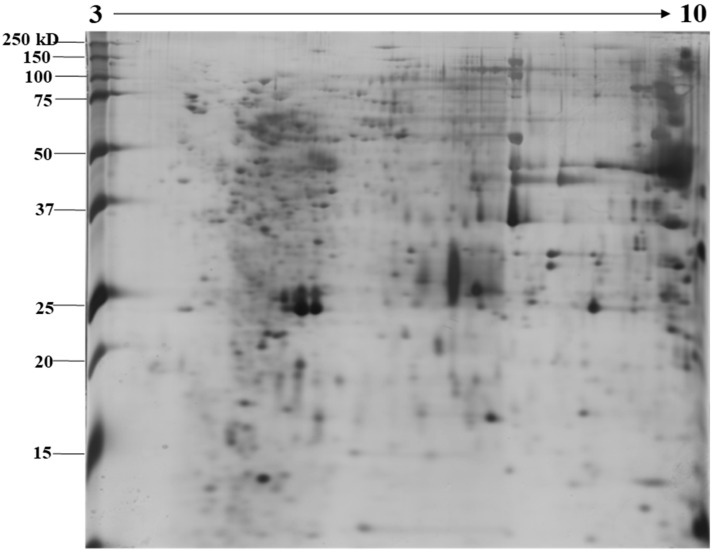
Image showing proteomic analysis of 2-DE of NnV. Representative 2-DE image of *N. nomurai* jellyfish venom (A). For the first dimension, 500 µg of proteins were resolved on an 18 cm, IPG dry strips (pH 3–10) and 12% SDS-PAGE gels were used to run second dimension. 2-DE gels were stained using the silver staining method and the Epson perfection V 700 photo scanner was used for scanning the stained gels. For statistical analysis, three independent replicate gels were run.

**Figure 2 toxins-11-00153-f002:**
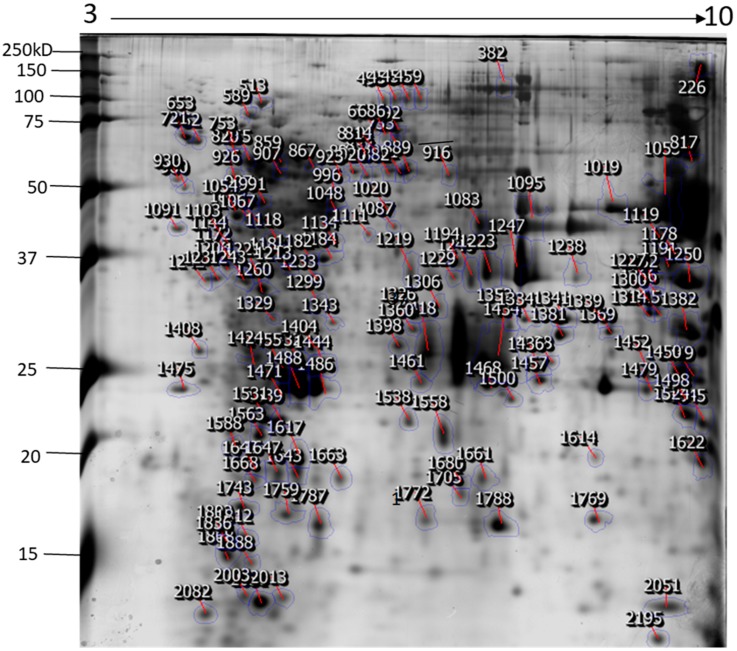
The representative 2-DE image of NnV proteins generated by Progenesis Same Spots software. Boundaries and arrows signify the position of differentially expressed proteins. Putative numbers were assigned to each protein spot.

**Figure 3 toxins-11-00153-f003:**
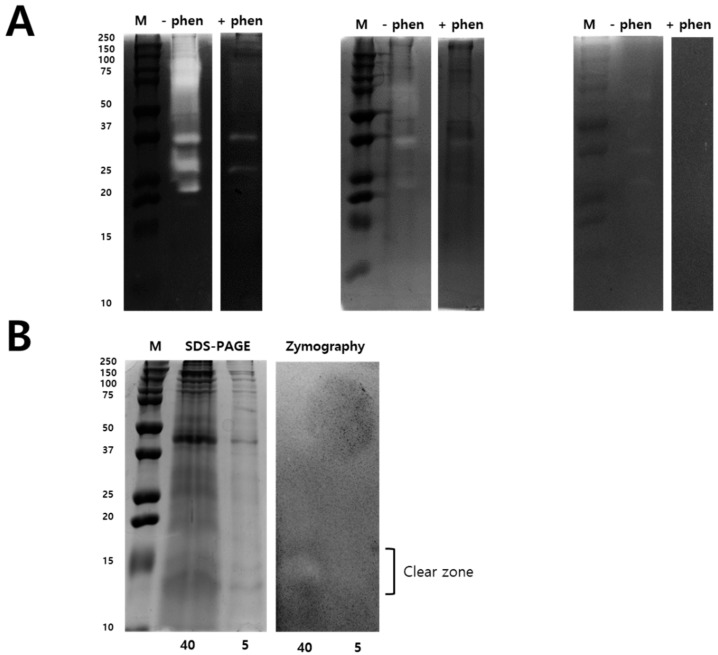
Zymography identify Metalloproteinase and PLA2 activity in NnV. Different types of zymography assays were performed to determining proteolytic activity of NnV (2 mg/mL) using gelatin, casein, and fibrin as a substrate and copolymerized in non-reducing SDS-PAGE (**A**). Zymography assays were performed in the presence of a metalloproteinase inhibitor (1,10-phenanthroline, 10 mM) during the enzymatic reaction time. Comparison of SDS-PAGE and zymography of NnV under non-reducing conditions (**B**). NnV proteins (40 mg and 5 mg of total protein) were run on a 12% SDS gel and gels were Coomassie stained. M: Protein molecular size marker. Clear zones in the gel indicate regions of proteolytic activity.

**Figure 4 toxins-11-00153-f004:**
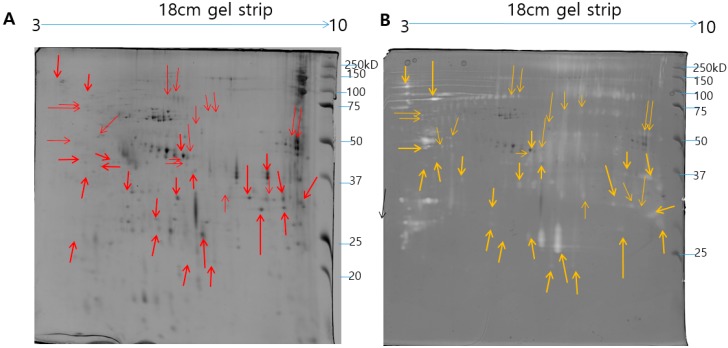
Comparison of 2-DE pattern and 2-DE gelatin zymogram of *N. nomurai* venom. The *N. nomurai* venom (500 µg) separated on 18 cm IPG dry strips (pH 3–10) in the first dimension then followed by second dimension (**A**). For the 2-DE gelatin zymography, the gels were copolymerized with gelatin and the second dimension was performed in a 12% SDS-PAGE gels under non-reducing conditions. Proteolytic activity is understood as clear zones of lysis against the dark background (**B**).

**Figure 5 toxins-11-00153-f005:**
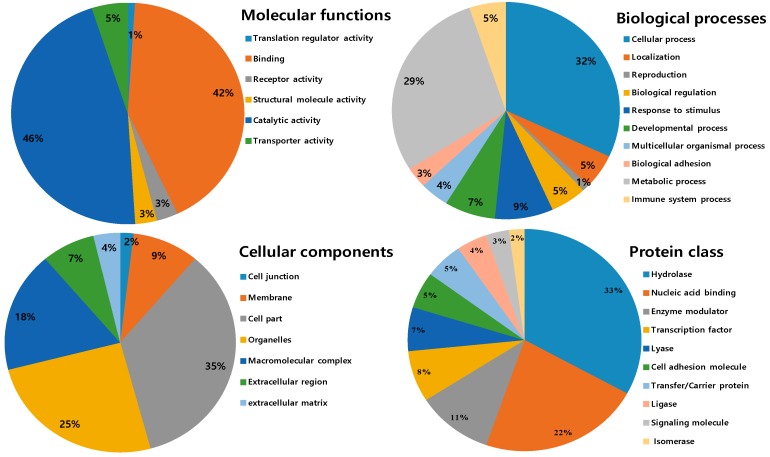
Gene ontology analysis of the identified venom proteins according to their molecular functions, biological processes, cell components, and protein classes.

**Figure 6 toxins-11-00153-f006:**
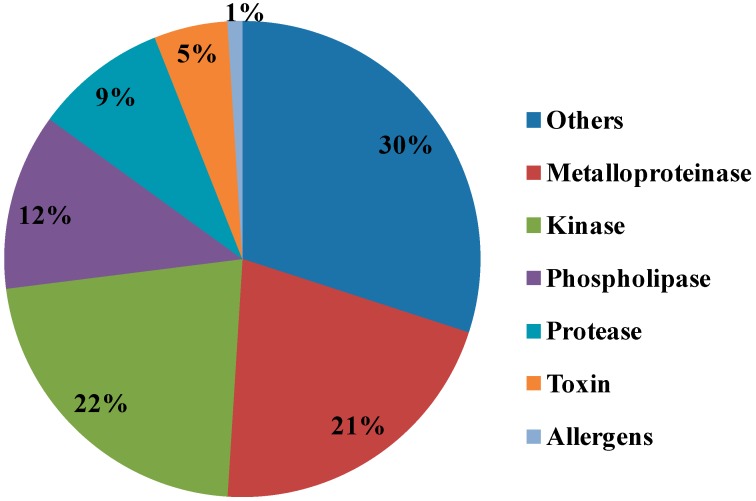
Comparison of relative composition of *N. nomurai* venom according to protein families by a proteomics approach. The pie chart is showing relative abundance of different protein families identified by MALDI/TOF/MS in NnV.

**Table 1 toxins-11-00153-t001:** Protein identified in *N. nomurai* venom by using MALDI/TOF/MS.

Spot No	Accession Number ^1^	Protein Name	Uniprot ID	Therotical MW/Pi ^2^	Organism	Matched Peptide ^3^	MOWSE Score	Biological Process
1450	P23897	Heat-stable enterotoxin recepter	GUC2C_RAT	123,468/6.4	*Rattus norvegicus*	8%	4851	Intracellular signal, transduction, regulation of cell proliferation
1053	P55128	RTX-I toxin determinant A from serotypes 1/9	RTXII_ACTPL	110,194/5.5	*Actinobacillus pleuropneumoniae*	9.20%	4636	Hemolysis in other organism, pathogenesis
1622	P15321	Hemolysin transporter, protein ShlB	HLYB_SERMA	61,917/9.2	*Serratia marcescens*	15.30%	3414	Hemolysis in other organism, pathogenesis protein transmembrane transport
1260	Q1W694	Phospholipase D LiSicTox-betaIDI	B1Q_LOXIN	34,831/7.6	*Loxosceles intermedia*	22.00%	154	Pathogenesis, Hemolysis in other organism, phospholipid catabolic process
1338	P39673	Allergen Mag (fragment)	MAG_DERFA	39,668/6.9	*Dermatophagoides farinae*	11.70%	320	
721	P55123	Leukotoxin	LKTA_PASSP	101,560/5.6	*Pasteurella haemo-lytica*-like sp. (strain 5943b)	8.40%	1331	Hemolysis in other organism, pathogenesis
1213	B2BS84	Putative Kunitz-type serine protease inhibitor	VKT_AUSLA	27,571/7.9	*Austrelaps labialis*	25.00%	7.14 × 10^2^	
1486	P0C845	Turripeptide Gsp9.1	C91_GEMSP	9290/9.1	*Gemmula speciosa*	26.80%	1802	
837	F5CPD3	Three-finger toxin MALT0044C	3SX4_MICAT	9398/8.4	*Micrurus altirostris*	54.10%	8534	Pathogenesis
1643	P81428	Venom prothrombin activator trocarin-D	FAXD_TROCA	51,407/8.1	*Tropidechis carinatus*	9.90%	294	Blood coagulation, envenomation resulting in positive regulation of blood coagulation in other organism
889	P55130	RTX-III toxin determinant A from serotype 2	RTX31_ACTPL	112,492/5.8	*Actinobacillus pleuropneumoniae*	13.00%	6.89 × 10^6^	Cytolysis, pathogenesis
930	O59824	ATP-dependent zinc metalloprotease YME1 homolog	YME1_SCHPO	78,219/8.5	*Schizosaccharomyces pombe* (strain 972/ATCC 24843)	6.20%	1335	proteolysis
1614	Q7S9D2	Pro-apoptotic serine protease nma111	NM111_NEUCR	113,312/5.7	*Neurospora crassa*	10.80%	30913	Apoptotic process
1182	P08026	Shiga-like toxin 1 subunit A	STXA_BPH19	34,800/9.6	*Enterobacteria phage* H19B	21%	2329	negative regulation of translation, pathogenesis
1450	Q2FZP2	Serine protease HtrA-like	HTRAL_STAA8	86,460/6.5	*Staphylococcus aureus* (strain NCTC 8325)	20.20%	876,777	
1647	E5AJX2	Snake venom serine protease nikobin	VSP_VIPNI	28,216/8.0	*Vipera nikolskii*	18.20%	425	
801	Q2QA02	Zinc metalloproteinase-disintegrin-like alternative name Snake venom metalloproteinase	VM3_CRODD	68,292/5.1	*Crotalus durissus durissus*	19.50%	57,907	
1233	Q40240	Major pollen allergen Lol p 5a	MPA5A_LOLPR	30,888/5.4	*Lolium perenne*	16.90%	707	type I hypersensitivity
1339	P54319	Phospholipase A-2-activating protein	PLAP_RAT	87,085/5.7	*Rattus norvegicus*	11.30%	1961	Inflammatory response
1343	P23636	Major serine/threonine-protein phosphatase PP2A_2 catalytic subunit	PP2A2_SCHPO	36,489/4.7	*Schizosaccharomyces pombe* (strain 972/ATCC 24843)	19.60%	2620	Cell division, signal transduction, mitotic nuclear division
1206	Q9TT93	A disintegrin and metallo-proteinase with thrombo-spondin motifs 4	ATS4_BOVIN	90,281/8.6	*Bos taurus*	11.90%	3.76 × 10^4^	Proteolysis, Angiogenesis
1500	P97570	85/88 kDa calcium-independent phospholipase A2	PLPL_RAT	89,556/6.7	*Rattus norvegicus*	12%	19,976	ATP dependent protein binding, chemotaxis, positive regulation of vasodilation
1315	P58459	A disintegrin and metalloproteinase with thrombospondin motifs 10	ATS10_MOUSE	121,087/8.4	*Mus musculus*	11.30%	40,176	Microfibrils assembly
1382	Q9R1V7	A disintegrin and metalloproteinase with thrombospondin motifs 23	ADA_23 MOUSE	91,548/7.9	*Mus musculus*	14%	2773	Cell adhesion
1461	Q9XWD6	Cell death abnormality protein 1	CED1_CAEEL	118,805/5.5	*Caenorhabditis elegans*	10.10%	6333	Programmed cell death, apoptotic cell death, receptor-mediated endocytosis,
1272	Q9T051	Phospholipase D gamma 2	PLDG2_ARATH	96,024/8.3	*Arabidopsis thaliana*	12.60%	11,703	Response to stress, membrane lipid metabolic process, phosphotidylcholine metabolic process
1334	Q10743	A disintegrin and metalloproteinase with thrombospondin motifs 10 fragment	ADA10_RAT	60,445/8.4	*Rattus norvegicus*	15.10%	5386	Negative regulation of cell adhesion, Notch signaling pathway, protein phosphorylation
1498	Q83XX3	ATP-dependent zinc metalloprotease FtsH	FTSH_OENOE	78,070/9.3	*Oenococcus oeni*	10.30%	447	Protein catabolic process
867	P78536	Disintegrin and metalloproteinase domain-containing protein 17	ADA17_HUMAN	93,022/5.5	*Homo sapiens*	10.70%	2397	Positive regulation of cell growth and cell migration, negative regulation of transforming growth factor beta receptor signaling
1434	A8XEZ1	Cell death abnormality protein 12	CED12_CAEBR	83,692/5.2	*Saccharomyces cerevisiae* (strain ATCC 204508/S288c)	9.00%	3371	Engulfment of apoptotic cell, cell migration, apoptotic process, phagocytosis, positive regulation of GTPase activity
1558	Q9S5Z2	ATP-dependent Clp protease ATP-binding subunit ClpE	PLPL8_MOUSE	87,382/9.3	*Mus musculus*	9.10%	2564	Cell death, arachidonic acid, secretion, phosphatidylcholine catabolic process
1418	Q8K1N1	Calcium-independent phospholipase A2-gamma	HTRAL_STAA8	86,460/6.5	*Staphylococcus aureus* (strain NCTC 8325)	20.20%	876,777	
1183	O93654	Tricorn protease-interacting factor F2	NAS8_CAEEL	46,096/5.8	*Caenorhabditis elegans*	13.40%	104	
1087	Q18439	Zinc metalloproteinase nas-8	CYM1_YEAST	112,181/6.0	*Saccharomyces cerevisiae* (strain ATCC 204508/S288c)	18.20%	383,182	Protein processing, proteolysis, 54

^1^ Accession numbers predicted by Swiss-Prot ^2^ Theoretical mass (MW) and Pi reported in Swiss-Prot ^3^ Percentage of amino acids sequence coverage of matched peptides for the identified proteins.

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
