# Peer review of "Proteomic Analysis of Novel Components of Nemopilema nomurai Jellyfish Venom: Deciphering the Mode of Action"

_toxins, 2019, doi:10.3390/toxins11030153_

Round 1

Author Response

To

The Editor

Toxin

Subject: Re-submission of edited manuscript: Proteomic analysis of novel components of Nemopilema nomurai jellyfish venom: Deciphering the Mode of Action

Dear Editor, 

I am writing to resubmit our manuscript “Proteomic analysis of novel components of Nemopilema nomurai jellyfish venom: Deciphering the Mode of Action” to be considered for publication in Toxins Journal. We would like to thank the editorial board and reviewers for the valuable comments. We have edited the manuscript according to the comments. Below we have addressed the changes that we have made as per the comments.

Answers to the reviewers’ comments

Reviewer 1

The discussion and conclusions of the manuscript must be rewritten. The discussion syntax seems confusing and there are no real conclusions from the results obtained in the text.

Authors must follow a format that place their findings, provide evidence of previous activity of the toxins found (they do something like that, but in an incorrect way) and discuss why they find such toxins in their proteome and what should be their function for the species of the study. In general, the article is interesting, but the presentation (style, figures, etc) needs changes

Answer:  We have modified the discussion and conclusion of the manuscript accordingly. We have also changed the figure style.

The information available to understand the zymography assays is incomplete, thus more information in the Figure 4 must be given in the picture and in the legend to make that information self-sufficient for better comprehension. For example, this information “Clear zones in the gel indicate regions of proteolytic activity” must be given in the caption of the Figure 4 instead of the section “Proteolytic activity assay” of Material and Methods.

Answer :  We agree with the reviewer’s comments and we have changed the material and method part as suggested.

Line 36: this sentence can be improved, maybe “N. nomurai is one of the giant and most dangerous jellyfish which belongs to Phylum Cnidaria, being their diagnostic feature the presence of stinging organelles called nematocysts, located mostly on the tentacles of jellyfish” [3-4], instead of

“N. nomurai is one of the giant and most dangerous jellyfish which belongs to Phylum Cnidaria and their determining characteristic is the presence of specialized cells known as nematocyte (cnidocyte) and stinging organelle nematocysts, located mostly on the tentacles of jellyfish [3-4]”.

Answer:  Thank you for the kind suggestions. We have edited the sentence as recommended

I recommend authors to change the title of the section “Modified zymography identify Metalloproteinase and PLA2 in NnV” by “Zymography assays demonstrate the enzymatic activity of the N. nomurai venom” because the enzymatic activity observed in the zymography cannot only ascribed to Metalloproteinase and PLA2. Other proteins found can also produce the effects observed.

Answer:  We agree with the comment and have edited the title of material method section accordingly.

I also recommend authors to improve Figure 3 avoiding the 3D-pie (with shadow also) and to improve the resolution and font of the Figure 5.

Answer:  We agree with the comments and have provided the improved figure 3. We have also changed the font of the figure 5 appropriately.

To check: Italicized latin names

Answer:  We have made the change accordingly

Line 43 remove “]” at the end of this sentence … body after NnV envenomation [9-10]].

Answer:  we have made the change accordingly

 Line “84” gymography instead zymography

Answer:  we have corrected the mistake accordingly.

Line “91” gymography

Answer:  we have corrected the mistake accordingly.

Line 89 “typos” in font and paragraph spacing: In fibrin zymography, weaker fibrinolytic activity was observed at 70 - 25 kDa

Answer

Line 174 “typos” in font and paragraph spacing: “in snake venom

Answer:  We have edited the sentence as per reviewer comments.

Line 175 “typos” in font and paragraph spacing: Additionally, recent studies revealed that

Answer:  We have edited the sentence as per reviewer comments.

--- All text from line 193 to line 196 is italicized

Answer:  We have made the change accordingly

Line 244 “typos” in font and paragraph spacing: by suppressing

Answer:

Line 262 “typos” Marlborough, MA, USA

Answer

This sentence is repeated in lines 59 and 250: “Till now the identification and functional characterization of its NnV components have been poorly described and beyond our knowledge”.

Answer:  We have edited the sentence as per reviewer comments.

 Figures are mentioned in the text as Figure 3.1, Figure 3.2 etc., however in the end of the document they are labeled just with Figure 3.

Answer: we have changed the figure labels as per reviewer comments.

Line 375: something is missing in the header of the Table 1.

Answer: we have agreed with the reviewer’s comments and modified the table 1.

Reviewer 2 Report

Jellyfish stings produce a variety of envenomation symptoms and signs on the human body, caused by protein/peptide toxins which in some cases can have life-threatening effects.  Therefore, comprehensive proteomic/genomic approaches applied to the study of these organisms, combined with in vivo/in vitro biological evaluation of such toxins, represent an invaluable tool for understanding their mechanism of action and also for developing strategies aimed to treat jellyfish stings. The manuscript entitled "Proteomic analysis of novel components of Nemopilema nomurai jellyfish venom: Deciphering the Mode of Action" describes the first  proteomic analysis of the jellyfish Nemopilema numurai, comprising 2-D electrophoresis combined with in-gel digestion and peptide mass fingerprinting characterization by  MALDI-TOF-MS, complemented with zymography techniques for the detection of proteolytic toxins. The authors identified a large number of proteins including some toxins, which could be responsible for the envenomation signs observed after a jellyfish sting.  The manuscript is organized and concise. Its content is within the scope of the journal, and it should be interesting for a large number of readers from different research areas related to toxins.

I have some comments to be addressed:

The authors used a Peptide mass fingerprinting methodology for protein identification. However, the identification can be difficult if the sample is composed of a mixture and not of a single protein, therefore needing an additional MS/MS approach.  Figure 2 shows many spots very close each other suggesting that there may be protein mixtures in some of them.  I would ask the author how they addressed this problem. Did the find protein mixtures or every spot was composed of a single protein? How did they proceed for the identification of proteins in mixtures? What criteria (detailed) were used ?  I think more details about the parameter values should be provided in section 5.6 (lines 332-334), regarding the identification.

Regarding the writing of the manuscript, I have some suggestions:

-line 23: the name of the species should be written in italics

-lines 10 and 119: “NnV ingredients”, to change it for “NnV components”.

-line9:   “undetermined”, to change it for “unknown”

-line 33: “From the last decades”, to change it for “Over recent decades”

-line 67: high homology, to change it for high sequence similarity. 

-line 68: is comprised of, to change it for is composed of

-line 70: disintigrin. Do the authors mean disintegrin? 

-line 90: were disappered, to change it for disappeared

-line 115: “turn”, to change it for “has turned”. 

-line 115: “give”, to change for  “gives”

-line 133. “were”, to change it for “was”

-line 134: “we have analyzed”, change it for “we have found”

-line 136-137: please rewrite the phrase “are the primary constituents of enzymatic toxins of various metalloproteinases”. The meaning is not clear.

-line 184: “can targeting”, change it for “can target”

-line 200: “it can be invented”, change it for “it can be inferred”

-line 251: “ and beyond our knowledge”, change it for “and remain beyond our knowledge”

-line 250: “its” should be removed

-line 251: “Best of our..”, change it for “To the best of our…”

Author Response

Reviewer 2

The authors used a Peptide mass fingerprinting methodology for protein identification. However, the identification can be difficult if the sample is composed of a mixture and not of a single protein, therefore needing an additional MS/MS approach.  Figure 2 shows many spots very close each other suggesting that there may be protein mixtures in some of them.  I would ask the author how they addressed this problem. Did the find protein mixtures or every spot was composed of a single protein? How did they proceed for the identification of proteins in mixtures? What criteria (detailed) were used ?  I think more details about the parameter values should be provided in section 5.6 (lines 332-334), regarding the identification.

Answer: We have precisely picked the single spot from the gel but while identification we got several proteins. The proteins with significant Mowse were considered andwe have already mentioned the parameters in the material and method section.

line 23: the name of the species should be written in italics

Answer:  We have made the change accordingly.

lines 10 and 119: “NnV ingredients”, to change it for “NnV components”.

Answer:  We have made the change accordingly.

line9:   “undetermined”, to change it for “unknown”

Answer:  We have made the change accordingly.

-line 33: “From the last decades”, to change it for “Over recent decades”

Answer:  We have made the change accordingly.

line 67: high homology, to change it for high sequence similarity. 

Answer:  We have made the change accordingly.

-line 68: is comprised of, to change it for is composed of

Answer:  We have made the change accordingly.

-

line 70: disintigrin. Do the authors mean disintegrin? 

Answer:  Yes, it is disintegrin.

line 90: were disappered, to change it for disappeared

Answer:  We have made the change accordingly.

-line 115: “turn”, to change it for “has turned”.

Answer:  We have made the change accordingly.

-line 115: “give”, to change for  “gives”

Answer:  We have made the change accordingly.

 line 133. “were”, to change it for “was”

Answer:  We have made the change accordingly.

line 134: “we have analyzed”, change it for “we have found”

Answer:  We have made the change accordingly.

-line 136-137: please rewrite the phrase “are the primary constituents of enzymatic toxins of various metalloproteinases”. The meaning is not clear.

Answer:  We have changed the phrase.

-line 184: “can targeting”, change it for “can target”

Answer:  We have made the change accordingly.

line 200: “it can be invented”, change it for “it can be inferred”

Answer:  We have made the change accordingly.

line 251: “ and beyond our knowledge”, change it for “and remain beyond our knowledge

Answer:  We have made the change accordingly.

line 250: “its” should be removed

Answer:  We have made the change accordingly.

Round 2

Reviewer 1 Report

Answer to the authors round 2

In general, this study is interesting for me and I would like to see published the manuscript. However, the writing and discussion must be improved

Line 41. Maybe better: N. nomurai known as the giant jellyfish is one of the most dangerous species belonging to the Phylum Cnidaria…than “N. nomurai is one of the giant and most dangerous jellyfish which belongs to Phylum Cnidaria”

Line 63-64. Change “Till now the identification and functional characterization of NnV components have been poorly described yet, and they are still beyond our knowledge. Our present study has successfully demonstrated the proteomic characterization of NnV…” by “…The identification and functional characterization of NnV…” components have been poorly characterized. Herein we described the proteomic profile of the NnV…”

line 69 Change “In this study, a total of 150 proteins identified from the nematocysts of NnV, includingg toxins” by “In this study, a total of 150 proteins identified from the nematocysts of NnV, including some toxins.

---In this sentence the last idea “…and another distinct type of proteins which are substantial in nematocyst and nematocyte generation (ref???)”  together with the following idea (see line 71) “…Interestingly, the identified toxins from N. nomurai jellyfish have shown high 71 sequence similarity with those of other venomous and poisonous animals” should be moved to the discussion and include a proper reference.

Line 84-“Confusing”- see “Modified zymography identify Metalloproteinase and PLA2 in NnV” maybe a typo

Line 87 authors used the adjective “extreme” to characterize the enzymatic activity of the NnV, but I should ask extreme compared to what? Maybe authors can use another adjective, or jus say higher enzymatic activity compared to …such species.

Line 89 Change “…Majority of the gelatinolytic activities…” by “The gelatinolytic activities of NnV…” and place the values obtained for the enzymatic activity tests. Authors also must reference the proper Figures (e.g Figure 4)

Line 132. Authors said: “… metalloproteinase are the second most 132 abundant components of NnV…” this is not real at all, since this study doesn´t evaluate the expression of such toxins at a quantitative level, so maybe should authors can clarify this idea.

Line 139. Authors said that in previous work they found a cytotoxicity effect produced by NnV “…which might play an essential role in inducing cytotoxicity…” but where? Better to place the complete idea to make more comprehensive this section.

Line 140-141 Authors said: “Therefore, Metalloproteinases in the venom of jellyfish Nemopilema nomurai might be responsible for swelling, inflammation, and dermonecrosis” Firstly, how I would like to know if the author previously evaluated the NnV in some models that allowed them to arrive to these conclusions, where is/are the references. Moreover, even if they can ascribe such symptom only to metalloproteases affects, if they have not evaluated any NnV components separated? I think this idea must be improved.

Line 142. Italice “in vivo and in vitro”

Line 137- to line 145. Author should consider moving this paragraph to the introduction (maybe)

Line 146- typo change “phospholipsaes” by phospholipases”

Line 155- authors must take care when say …”third abundant” since they are not performing a quantitative analysis here, should be improved

Line 168 –“…NnV might be responsible for pathogenesis during NnV envenomation..”---I think authors should clarify what kind of pathogenesis are produced by serine proteases. Also, I recommend authors to avoid this kind of ideas that don’t led the reader to understand the role of some venom components, instead can create more confusion about if the authors are evaluating the NnV or isolated venom component such as serine proteases or metalloproteases.

Line 174- typo in font format “…in snake venom [48-50]. Additionally, recent studies revealed that..”

Line 239. “…Metabolomic or metabolic…”

Line 251- Conclusion: authors should improve the conclusions, some information of the main finding can be added, like the composition of NnV, and how such toxins correlates with symptoms observed, or how this component can provide ecological advantage to the studied species.

Line 390- Figure 2. I think the label in the Figure can be edited, or changed by the accession number of the toxins found.

Author Response

Reviewer 1

Line 41. Maybe better: N. nomurai known as the giant jellyfish is one of the most dangerous species belonging to the Phylum Cnidaria…than “N. nomurai is one of the giant and most dangerous jellyfish which belongto Phylum Cnidaria”

Replay : we agreed with reviewer comments and we have made the change accordingly

Line 63-64. Change “Till now the identification and functional characterization of NnV components have been poorly described yet, and they are still beyond our knowledge. Our present study has successfully demonstrated the proteomic characterization of NnV…” by “…The identification and functional characterization of NnV…” components have been poorly characterized. Herein we described the proteomic profile of the NnV…

Replay : We have made the change accordingly

line 69 Change “In this study, a total of 150 proteins identified from the nematocysts of NnV, includingg toxins” by “In this study, a total of 150 proteins identified from the nematocysts of NnV, including some toxins.

---In this sentence the last idea “…and another distinct type of proteins which are substantial in nematocyst and nematocyte generation (ref???)”  together with the following idea (see line 71) “…Interestingly, the identified toxins from N. nomurai jellyfish have shown high 71 sequence similarity with those of other venomous and poisonous animals” should be moved to the discussion and include a proper reference.

Replay : We have made the change accordingly and we have mentioned the venomous proteins in the discussion with reference

Line 84-“Confusing”- see “Modified zymography identify Metalloproteinase and PLA2 in NnV” maybe a typo

Replay : We have made the change accordingly

Line 87 authors used the adjective “extreme” to characterize the enzymatic activity of the NnV, but I should ask extreme compared to what? Maybe authors can use another adjective, or jus say higher enzymatic activity compared to …such species.

Replay : We have made the change accordingly

Line 89 Change “…Majority of the gelatinolytic activities…” by “The gelatinolytic activities of NnV…” and place the values obtained for the enzymatic activity tests. Authors also must reference the proper Figures (e.g Figure 4)

Replay : We have made the change accordingly and we have already mentioned figure 4.

Line 132. Authors said: “… metalloproteinase are the second most 132 abundant components of NnV…” this is not real at all, since this study doesn´t evaluate the expression of such toxins at a quantitative level, so maybe should authors can clarify this idea.

Replay:  Here we are considering the toxin protein . Among the toxin proteins metalloproteinase are the second most abundant components of NnV.

 Line 139. Authors said that in previous work they found a cytotoxicity effect produced by NnV “…which might play an essential role in inducing cytotoxicity…” but where? Better to place the complete idea to make more comprehensive this section.

Replay :  We agreed with reviewer comments, we have made the change accordingly

Line 140-141 Authors said: “Therefore, Metalloproteinases in the venom of jellyfish Nemopilema nomurai might be responsible for swelling, inflammation, and dermonecrosis” Firstly, how I would like to know if the author previously evaluated the NnV in some models that allowed them to arrive to these conclusions, where is/are the references. Moreover, even if they can ascribe such symptom only to metalloproteases affects, if they have not evaluated any NnV components separated? I think this idea must be improved.

Replay: I agreed with reviewer comments but the dermal toxicity is well studied in our lab previously and I have given the reference.

Line 142. Italice “in vivo and in vitro”

Replay : We have made the change accordingly

Line 146- typo change “phospholipsaes” by phospholipases”

Replay : We have made the change accordingly

Line 155- authors must take care when say …”third abundant” since they are not performing a quantitative analysis here, should be improved

Replay : We agreed with reviewer comments and we have improved the sentence.

Line 174- typo in font format “…in snake venom [48-50]. Additionally, recent studies revealed that..

Replay : We have made the change accordingly

Line 239. “…Metabolomic or metabolic…”

Replay : We have made the change accordingly

Line 251- Conclusion: authors should improve the conclusions, some information of the main finding can be added, like the composition of NnV, and how such toxins correlates with symptoms observed, or how this component can provide ecological advantage to the studied species.

Replay : we have modified the conclusion accordingly

Line 390- Figure 2. I think the label in the Figure can be edited, or changed by the accession number of the toxins found.

Replay : Figure 2 is the image which is generated by progenesis software and it can not be changed into accession numbers.
